# Facility Location Problem in Differential Privacy Model Revisited

**Yunus Esencayi** [*]
SUNY at Buffalo
yunusese@buffalo.edu

**Marco Gaboardi**
Boston University
gaboardi@bu.edu

**Shi Li**
SUNY at Buffalo
shil@buffalo.edu

**Di Wang**
SUNY at Buffalo
dwang45@buffalo.edu

## Abstract

In this paper we study the uncapacitated facility location problem in the model of differential privacy (DP) with uniform facility cost. Specifically, we first show that, under the *hierarchically well-separated tree (HST) metrics* and the *super-set output setting* that was introduced in [8], there is an $\epsilon$-DP algorithm that achieves an $O(\frac{1}{\epsilon})$ (expected multiplicative) approximation ratio; this implies an $O(\frac{\log n}{\epsilon})$ approximation ratio for the general metric case, where $n$ is the size of the input metric. These bounds improve the best-known results given by [8]. In particular, our approximation ratio for HST-metrics is independent of $n$, and the ratio for general metrics is independent of the aspect ratio of the input metric.

On the negative side, we show that the approximation ratio of any $\epsilon$-DP algorithm is lower bounded by $\Omega(\frac{1}{\sqrt{\epsilon}})$, even for instances on HST metrics with uniform facility cost, under the super-set output setting. The lower bound shows that the dependence of the approximation ratio for HST metrics on $\epsilon$ can not be removed or greatly improved. Our novel methods and techniques for both the upper and lower bound may find additional applications.

## 1 Introduction

The facility location problem is one of the most fundamental problems in combinatorial optimization and has a wide range of applications such as plant or warehouse location problems and network design problems, also it is closely related to clustering problems such as $k$-median, where one typically seeks to partition a set of data points, which themselves find applications in data mining, machine learning, and bioinformatics [1, 13, 4]. Due to its versatility, the problem has been studied by both operations research and computer science communities [20, 19, 16, 15, 1, 13, 4]. Formally, it can be defined as following.

**Definition 1** (Uniform Facility Location Problem (Uniform-FL))**.** The input to the Uniform Facility Location (Uniform-FL) problem is a tuple $(V, d, f, \vec{N})$, where $(V, d)$ is a $n$-point discrete metric, $f \in \mathbb{R}_{\geq 0}$ is the facility cost, and $\vec{N} = (N_v)_{v \in V} \in \mathbb{Z}_{\geq 0}^V$ gives the number of clients in each location $v \in V$. The goal of the problem is to find a set of facility locations $S \subseteq V$ which minimize the following, where $d(v, S) = \min_{s \in S} d(v, s)$,

$$\min_{S \subseteq V} \text{cost}_d(S; \vec{N}) := |S| \cdot f + \sum_{v \in V} N_v d(v, S). \tag{1}$$

The first term of (1) is called the *facility cost* and the second term is called the *connection cost*.

---

[*]Authors are alphabetically ordered.

Throughout the paper, we shall simply use **UFL** to refer to Uniform-FL. Although the problem has been studied quite well in recent years, there is some privacy issue on the locations of the clients. Consider the following scenario: One client may get worried that the other clients may be able to obtain some information on her location by colluding and exchanging their information. As a commonly-accepted approach for preserving privacy, Differential Privacy (DP) [5] provides provable protection against identification and is resilient to arbitrary auxiliary information that might be available to attackers.

However, under the $\epsilon$-DP model, Gupta et al. [8] recently showed that it is impossible to achieve a useful multiplicative approximation ratio of the facility location problem. Specifically, they showed that any 1-DP algorithm for UFL under general metric that outputs the set of open facilities must have a (multiplicative) approximation ratio of $\Omega(\sqrt{n})$ which negatively shows that UFL in DP model is useless. Thus one needs to consider some relaxed settings in order to address the issue.

In the same paper [8] the authors showed that, under the following setting, an $O(\frac{\log^2 n \log^2 \Delta}{\epsilon})$ approximation ratio under the $\epsilon$-DP model is possible, where $\Delta = \max_{u,v \in V} d(u,v)$ is the diameter of the input metric. In the setting, the output is a set $R \subseteq V$, which is a *super-set* of the set of open facilities. Then every client sees the output $R$ and chooses to connect to its nearest facility in $R$. The the actual set $S$ of open facilities, is the facilities in $R$ with at least one connecting client. **Thus, in this model, a client will only know its own service facility, instead of the set of open facilities.** We call this setting the *super-set output setting*. Roughly speaking, under the $\epsilon$-DP model, one can not well distinguish between if there is 0 or 1 client at some location $v$. If $v$ is far away from all the other locations, then having one client at $v$ will force the algorithm to open $v$ and thus will reveal information about the existence of the client at $v$. This is how the lower bound in [8] was established. By using the super-set output setting, the algorithm can always output $v$ and thus does not reveal much information about the client. If there is no client at $v$ then $v$ will not be open.

In this paper we further study the UFL problem in the $\epsilon$-DP model with the super-set output setting by [8] we address the following questions.

> For the UFL problem under the $\epsilon$-DP model and the super-set output setting, can we do better than the results in [8] in terms of the approximation ratio? Also, what is the lower bound of the approximation ratio in the same setting?

We make progresses on both problems. Our contributions can be summarized as the followings.

- We show that under the so called *Hierarchical-Well-Separated-Tree (HST)* metrics, there is an algorithm that achieves $O(\frac{1}{\epsilon})$ approximation ratio. By using the classic FRT tree embedding technique of [6], we can achieve $O(\frac{\log n}{\epsilon})$ approximation ratio for any metrics, under the $\epsilon$-DP model and the super-set output setting. These factors respectively improve upon a factor of $O(\log n \log^2 \Delta)$ in [8] for HST and general metrics. Thus, for HST-metrics, our approximation only depends on $\epsilon$. For general metrics, our result removed the poly-logarithmic dependence on $\Delta$ in [8].

- On the negative side, we show that the approximation ratio under $\epsilon$-DP model is lower bounded by $\Omega(\frac{1}{\sqrt{\epsilon}})$ even if the metric is a star (which is a special case of a HST). This shows that the dependence on $\epsilon$ is unavoidable and can not been improved greatly.

**Related Work**   The work which is the most related to this paper is [8], where the author first studied the problem. Nissim et al. [18] study an abstract mechanism design model where DP is used to design approximately optimal mechanism, and they use facility location as one of their key examples. The UFL problem has close connection to $k$-median clustering and submodular optimization, whose DP versions have been studied before such as [17, 3, 7, 8, 2]. However, their methods cannot be used in our problem. There are many papers study other combinatorial optimization problems in DP model such as [9, 10, 11, 12, 8]. Finally, we remark that the setting we considered in the paper is closely related to the *Joint Differential Privacy Model* that was introduced in [14]. We leave the details to the full version of the paper.

## 2   Preliminaries

Given a data universe $V$ and a dataset $D = \{v_1, \cdots, v_N\} \in V^N$ where each record $v_i$ belongs to an individual $i$ whom we refer as a client in this paper. Let $\mathcal{A} : V^N \mapsto \mathcal{S}$ be an algorithm on $D$ and produce an output in $\mathcal{S}$. Let $D_{-i}$ denote the dataset $D$ without entry of the $i$-th client. Also $(v'_i, D_{-i})$ denote the dataset by adding $v'_i$ to $D_{-i}$.

**Definition 2** (Differential Privacy [5]). A randomized algorithm $\mathcal{A}$ is $\epsilon$-differentially private (DP) if for any client $i \in [N]$, any two possible data entries $v_i, v_i' \in V$, any dataset $D_{-i} \in V^{N-1}$ and for all events $\mathcal{T} \subseteq \mathcal{S}$ in the output space of $\mathcal{A}$, we have $\Pr[\mathcal{A}(v_i, D_{-i}) \in \mathcal{T}] \leq e^\epsilon \Pr[\mathcal{A}(v_i', D_{-i}) \in \mathcal{T}]$.

For the UFL problem, instead of using a set $D$ of clients as input, it is more convenient for us to use a vector $\vec{N} = (N_v)_{v \in V} \in \mathbb{Z}_{\geq 0}^V$, where $N_v$ indicates the number of clients at location $v$. Then the $\epsilon$-DP requires that for any input vectors $\vec{N}$ and $\vec{N}'$ with $|\vec{N} - \vec{N}'|_1 = 1$ and any event $\mathcal{T} \subseteq \mathcal{S}$, we have $\Pr[\mathcal{A}(\vec{N}) \in \mathcal{T}] \leq e^\epsilon \Pr[\mathcal{A}(\vec{N}') \in \mathcal{T}]$.

In the super-set output setting for the UFL problem, the output of an algorithm is a set $R \subseteq V$ of potential open facilities. Then, every client, or equivalently, every location $v$ with $N_v \geq 1$, will be connected to the nearest location in $R$ under some given metric (in our algorithm, we use the HST tree metric). Then the actual set $S$ of open facilities is the set of locations in $R$ with at least 1 connected client. Notice that the connection cost of $S$ will be the same as that of $R$; but the facility cost might be much smaller than that of $R$. This is why the super-set output setting may help in getting good approximation ratios.

**Throughout the paper, approximation ratio of an algorithm $\mathcal{A}$ is the *expected multiplicative approximation ratio*, which is the expected cost of the solution given by the algorithm, divided by the cost of the optimum solution**, *i.e.,* $\frac{\mathbb{E}\mathrm{cost}_d(\mathcal{A}(\vec{N}); \vec{N})}{\min_{S \subseteq V} \mathrm{cost}_d(S; \vec{N})}$, where the expectation is over the randomness of $\mathcal{A}$.

**Organization**  In Section 3, we show how to reduce UFL on general metrics to that on HST metrics, while losing a factor of $O(\log n)$ in the approximation ratio. In Section 4, we give our $\epsilon$-DP $O(1/\epsilon)$-approximation for UFL under the super-set output setting. Finally in Section 5, we prove our $\Omega(1/\sqrt{\epsilon})$-lower bound on the approximation ratio for the same setting. All missing proofs will be deferred to the full version of the paper.

# 3 Reducing General Metrics to Hierarchically Well-Separated Tree Metrics

The classic result of Fakcharoenphol, Rao and Talwar (FRT) [6] shows that any metric on $n$ points can be embedded into a distribution of metrics induced by *hierarchically well-separated trees* with distortion $O(\log n)$. As in [8], this tree-embedding result is our starting point for our DP algorithm for uniform UFL. To apply the technique, we first define what is a hierarchically well-separated tree.

**Definition 3.** For any real number $\lambda > 1$, an integer $L \geq 1$, a $\lambda$-Hierarchically Well-Separated tree ($\lambda$-HST) of depth $L$ is an edge-weighted rooted tree $T$ satisfying the following properties:

(3a) Every root-to-leaf path in $T$ has exactly $L$ edges.

(3b) If we define the level of a vertex $v$ in $T$ to be $L$ minus the number of edges in the unique root-to-$v$ path in $T$, then an edge between two vertices of level $\ell$ and $\ell + 1$ has weight $\lambda^\ell$.

Given a $\lambda$-HST $T$, we shall always use $V_T$ to denote its vertex set. For a vertex $v \in V_T$, we let $\ell_T(v)$ denote the *level of $v$* using the definition in (3b). Thus, the root $r$ of $T$ has level $\ell_T(r) = L$ and every leaf $v \in T$ has level $\ell_T(v) = 0$. For every $u, v \in V_T$, define $d_T(u, v)$ be the total weight of edges in the unique path from $u$ to $v$ in $T$. So $(V_T, d_T)$ is a metric. With the definitions, we have:

**Fact 4.** Let $u \in V_T$ be a non-leaf of $T$ and $v \neq u$ be a descendant of $u$, then

$$\lambda^{\ell_T(u)-1} \leq d_T(u, v) \leq \frac{\lambda^{\ell_T(u)}-1}{\lambda-1} \leq \frac{\lambda^{\ell_T(u)}}{\lambda-1}.$$

We say a metric $(V, d)$ is a $\lambda$-HST metric for some $\lambda > 1$ if there exists a $\lambda$-HST $T$ with leaves being $V$ such that $(V, d) \equiv (V, d_T|_V)$, where $d_T|_V$ is the function $d_T$ restricted to pairs in $V$. Throughout the paper, we guarantee that if a metric is a $\lambda$-HST metric, the correspondent $\lambda$-HST $T$ is given. We give the formal description of the FRT result as well how to apply it to reduce UFL on general metrics to that on $O(1)$-HST metrics in the full version of the paper.

Specifically, we shall prove the following theorem:

**Theorem 5.** Let $\lambda > 1$ be any absolute constant. If there exists an efficient $\epsilon$-DP $\alpha_{\mathrm{tree}}(n, \epsilon)$-approximation algorithm $\mathcal{A}$ for UFL on $\lambda$-HST's under the super-set output setting, then there exists an efficient $\epsilon$-DP $O(\log n) \cdot \alpha_{\mathrm{tree}}(n, \epsilon)$-approximation algorithm for UFL on general metrics under the same setting.

In Section 4, we shall show that it is possible to make $\alpha_{\text{tree}}(n, \epsilon) = O(1/\epsilon)$:

**Theorem 6.** For every small enough absolute constant $\lambda > 1$, there is an efficient $\epsilon$-DP $O(1/\epsilon)$-approximation algorithm for UFL on $\lambda$-HST metrics under the super-set output setting.

Combining Theorems 5 and 6 will give our main theorem.

**Theorem 7** (Main Theorem). Given any UFL tuple $(V, d, f, \vec{N})$ where $|V| = n$ and $\epsilon > 0$, there is an efficient $\epsilon$-DP algorithm $\mathcal{A}$ in the super-set output setting achieving an approximation ratio of $O(\frac{\log n}{\epsilon})$.

## 4 $\epsilon$-DP Algorithm with $O(1/\epsilon)$ Approximation Ratio for HST Metrics

In this section, we prove Theorem 6. Let $\lambda \in (1, 2)$ be any absolute constant and let $\eta = \sqrt{\lambda}$. We prove the theorem for this fixed $\lambda$. So we are given a $\lambda$-HST $T$ with leaves being $V$. Our goal is to design an $\epsilon$-DP $\left(\alpha_{\text{tree}} = O(1/\epsilon)\right)$-approximation algorithm for UFL instances on the metric $(V, d_T)$. Our input vector is $\vec{N} = (N_v)_{v \in V}$, where $N_v \in \mathbb{Z}_{\geq 0}$ is the number of clients at the location $v \in V$.

### 4.1 Useful Definitions and Tools

Before describing our algorithm, we introduce some useful definitions and tools. Recall that $V_T$ is the set of vertices in $T$ and $V \subseteq V_T$ is the set of leaves. Since we are dealing with a fixed $T$ in this section, we shall use $\ell(v)$ for $\ell_T(v)$. Given any $u \in V_T$, we use $T_u$ to denote the sub-tree of $T$ rooted at $u$. Let $L \geq 1$ be the depth of $T$; we assume $L \geq \log_\lambda(\epsilon f)$.[2] We use $L' = \max\{0, \lceil \log_\lambda(\epsilon f) \rceil\} \leq L$ to denote the smallest non-negative integer $\ell$ such that $\lambda^\ell \geq \epsilon f$.

We extend the definition of $N_u$'s to non-leaves $u$ of $T$: For every $u \in V_T \setminus V$, let $N_u = \sum_{v \in T_u \cap V} N_v$ to be the total number of clients in the tree $T_u$.

We can assume that facilities can be built at any location $v \in V_T$ (instead of only at leaves $V$): On one hand, this assumption enriches the set of valid solutions and thus only decreases the optimum cost. On the other hand, for any $u \in V_T$ with an open facility, we can move the facility to any leaf $v$ in $T_u$. Then for any leaf $v' \in V$, it is the case that $d(v', v) \leq 2d(v', u)$. Thus moving facilities from $V_T \setminus V$ to $V$ only incurs a factor of 2 in the connection cost.

An important function that will be used throughout this section is the following set of *minimal vertices*:

**Definition 8.** For a set $M \subseteq T$ of vertices in $T$, let

$$\text{min-set}(M) := \{u \in M : \forall v \in T_u \setminus \{u\}, v \notin M\}.$$

For every $v$, let we define $B_v := \min\{f, N_v \lambda^{\ell(v)}\}$. This can be viewed as a lower bound on the cost incurred inside the tree $T_v$, as can be seen from the following claim:

**Claim 9.** Let $V' \subseteq V_T$ be a subset of vertices that does not contain an ancestor-descendant pair[3]. Then we have $\text{opt} \geq \sum_{v \in V'} B_v$.

### 4.2 Base Algorithm for UFL without Privacy Guarantee

Before describing the $\epsilon$-DP algorithm, we first give a base algorithm (Algorithm 1) without any privacy guarantee as the starting point of our algorithmic design. The algorithm gives an approximation ratio of $O(1/\epsilon)$; however, it is fairly simple to see that by making a small parameter change, we can achieve $O(1)$-approximation ratio. We choose to present the algorithm with $O(1/\epsilon)$-ratio only to make it closer to our final algorithm (Algorithm 2), which is simply the noise-version of the base algorithm. The noise makes the algorithm $\epsilon$-DP, while only incurring a small loss in the approximation ratio.

Recall that we are considering the super-set output setting, where we return a set $R$ of facilities, but only open a set $S \subseteq R$ of facilities using the following *closest-facility rule*: We connect every client to its nearest facility in $R$, then the set $S \subseteq R$ of open facilities is the set of facilities in $R$ with at least 1 connected client.

**Algorithm 1** UFL-tree-base($\epsilon$)

---

1: $L' \leftarrow \max\{0, \lceil \log_\lambda(\epsilon f) \rceil\}$

2: Let $M \leftarrow \left\{ v \in V_T : \ell(v) \geq L' \text{ or } N_v \cdot \lambda^{\ell(v)} \geq f \right\}$ be the set of marked vertices

3: $R \leftarrow \text{min-set}(M)$

4: **return** $R$ but only open $S \subseteq R$ using the closest-facility rule.

---

In the base algorithm, $M$ is the set of *marked* vertices in $T$ and we call vertices not in $M$ *unmarked*. All vertices at levels $[L', L]$ are marked. Notice that there is a monotonicity property among vertices in $V_T$: for two vertices $u, v \in V_T$ with $u$ being an ancestor of $v$, $v$ is marked implies that $u$ is marked. Due to this property, we call an unmarked vertex $v \in V_T$ *maximal-unmarked* if its parent is marked. Similarly, we call a marked vertex $v \in V_T$ *minimal-marked* if all its children are unmarked (this is the case if $v$ is a leaf). So $R$ is the set of minimal-marked vertices. Notice one difference between our algorithm and that of [8]: we only return minimal-marked vertices, while [8] returns all marked ones. This is one place where we can save a logarithmic factor, which requires more careful analysis.

We bound the facility and connection cost of the solution $S$ given by Algorithm 1 respectively. Indeed, for the facility cost, we prove some stronger statement. Define $V^\circ = \{u \in V_t : N_u \geq 1\}$ be the set of vertices $u$ with at least 1 client in $T_u$. We prove

**Claim 10.** $S \subseteq \text{min-set}(V^\circ \cap M)$.

The stronger statement we prove about the facility cost of the solution $S$ is the following:

**Lemma 11.** $|\text{min-set}(V^\circ \cap M)| \cdot f \leq (1 + 1/\epsilon)\text{opt}$.

Notice that Claim 10 and Lemma 11 imply that $|S| \cdot f \leq O(1 + 1/\epsilon)\text{opt}$.

Now we switch gear to consider the connection cost of the solution $S$ and prove:

**Lemma 12.** The connection cost of $S$ given by the base algorithm is at most $O(1)\text{opt}$.

### 4.3 Guaranteeing $\epsilon$-DP by Adding Noises

In this section, we describe the final algorithm (Algorithm 2) that achieves $\epsilon$-DP without sacrificing the order of the approximation ratio. Recall that $\eta = \sqrt{\lambda}$.

---

**Algorithm 2** DP-UFL-tree($\epsilon$)

---

1: $L' \leftarrow \max\{0, \lceil \log_\lambda(\epsilon f) \rceil\}$

2: for every $v \in V_T$ with $\ell(v) < L'$, define $\tilde{N}_v := N_v + \text{Lap}\left( \frac{f}{c\eta^{L'+\ell(v)}} \right)$, where $c = \frac{\eta-1}{\eta^2}$.

3: Let $M \leftarrow \left\{ v \in V_T : \ell(v) \geq L' \text{ or } \tilde{N}_v \cdot \lambda^{\ell(v)} \geq f \right\}$ be the set of marked vertices

4: $R \leftarrow \text{min-set}(M)$

5: **return** $R$ but only open $S \subseteq R$ using the closest-facility rule.

---

We give some intuitions on how we choose the noises in Step 1 of the Algorithm. Let us travel through the tree from level $L'$ down to level $0$. Then the Laplacian parameter, which corresponds to the magnitude of the Laplacian noise, goes up by factors of $\eta$. This scaling factor is carefully chosen to guarantee two properties. First the noise should go up exponentially so that the DP parameter only depends on the noise on the highest level, i.e, level $L'$. Second, $\eta$ is smaller than the distance scaling factor $\lambda = \eta^2$. Though the noises are getting bigger as we travel down the tree, their effects are getting smaller since they do not grow fast enough. Then essentially, the effect of the noises is only on levels near $L'$.

**Lemma 13.** Algorithm 2 satisfies $\epsilon$-DP property.

### 4.4 Increase of cost due to the noises

We shall analyze how the noise affects the facility and connection costs. Let $M^0, R^0$ and $S^0$ (resp. $M^1, R^1$ and $S^1$) be the $M, R$ and $S$ generated by Algorithm 1 (resp. Algorithm 2). In the proof,

we shall also consider running Algorithm 1 with input vector being $2\vec{N}$ instead of $\vec{N}$. Let $M'^0, R'^0$ and $S'^0$ be the $M, R$ and $S$ generated by Algorithm 1 when the input vector is $2\vec{N}$. Notice that the optimum solution for input vector $2\vec{N}$ is at most 2opt. Thus, Lemma 11 implies $|S'^0| \cdot f = O(1/\epsilon)$opt. Notice that $M^0, R^0, S^0, M'^0, R'^0$ and $S'^0$ are deterministic while $M^1, R^1$ and $S^1$ are randomized.

The lemmas we shall prove are the following:

**Lemma 14.** $\mathbb{E}\left[|S^1| \cdot f\right] \leq O(1/\epsilon) \cdot \text{opt}$.

**Lemma 15.** The expected connection cost of the solution $S^1$ is $O(1)$ times that of $S^0$.

Thus, combining the two lemmas, we have that the expected cost of the solution $S^1$ is at most $O(1/\epsilon)$opt, finishing the proof of Theorem 6. Indeed, we only lose an $O(1)$-factor for the connection cost as both factors in Lemma 11 and 15 are $O(1)$. We then prove the two lemmas separately.

### 4.4.1 Increase of facility costs due to the noise

In this section, we prove Lemma 14. A technical lemma we can prove is the following:

**Claim 16.** Let $M \subseteq V_T$ and $M' = M \cup \{v\}$ for some $v \in V_T \setminus M$, then exactly one of following three cases happens.

(16a) $\text{min-set}(M') = \text{min-set}(M)$.

(16b) $\text{min-set}(M') = \text{min-set}(M) \uplus \{v\}$.

(16c) $\text{min-set}(M') = \text{min-set}(M) \setminus \{u\} \cup \{v\}$, where $u \in \text{min-set}(M), v \notin \text{min-set}(M)$ and $v$ is a descendant of $u$.

*Proof of Lemma 14.* Recall that $V^\circ$ is the set of vertices $u$ with $N_u \geq 1$. We first focus on open facilities in $V^\circ$ in $S^1$. Claim 16 implies that adding one new element to $M$ will increase $|\text{min-set}(M)|$ by at most 1. Thus, we have

$$|\text{min-set}(M^1 \cap V^\circ)| - |\text{min-set}(M'^0 \cap V^\circ)| \leq |(M^1 \cap V^\circ) \setminus (M'^0 \cap V^\circ)|$$

$$= \left|\left\{u \in V^\circ : \ell(u) < L', 2N_u < \frac{f}{\lambda^{\ell(u)}} \leq \tilde{N}_u\right\}\right|.$$

We now bound the expectation of the above quantity. Let $U^*$ be the set of vertices $u \in V^\circ$ with $\ell(u) < L'$ and $N_u < \frac{f}{2\lambda^{\ell(u)}}$. Then for every $u \in U^*$, we have

$$\Pr[u \in M^1] = \Pr\left[N_v + \text{Lap}\left(\frac{f}{c\eta^{L'+\ell(u)}}\right) \geq \frac{f}{\lambda^{\ell(u)}}\right]$$

$$\leq \frac{1}{2}\exp\left(-\frac{f/(2\lambda^{\ell(u)})}{f/(c\eta^{L'+\ell(u)})}\right) = \frac{1}{2}\exp\left(-\frac{c\eta^{L'-\ell(u)}}{2}\right). \tag{2}$$

We bound $f$ times the sum of (2), over all $u \in U^*$. Notice that every $u \in V^\circ$ has $N_u \geq 1$. So we have $B_u = \min\left\{f, N_u\lambda^{\ell(u)}\right\} \geq \lambda^{\ell(u)}$ for every $u$ we are interested in. Then,

$$f \leq \frac{1}{\epsilon} \cdot \epsilon f \cdot \frac{B_u}{\lambda^{\ell(u)}} \leq \frac{1}{\epsilon} \cdot \lambda^{L'} \cdot \frac{B_u}{\lambda^{\ell(u)}} = \frac{B_u}{\epsilon} \cdot \eta^{2(L'-\ell(u))}. \tag{3}$$

The last inequality comes from $\epsilon f \leq \lambda^{L'}$. The equality used that $\lambda = \eta^2$.

We group the $u$'s according to $\ell(u)$. For each level $\ell \in [0, L'-1]$, we have

$$\frac{f}{2}\sum_{u \in U^*:\ell(u)=\ell}\exp\left(-\frac{c\eta^{L'-\ell(u)}}{2}\right) \leq \frac{1}{2\epsilon}\eta^{2(L'-\ell)}\exp\left(-\frac{c\eta^{L'-\ell}}{2}\right)\sum_{u \text{ as before}}B_u \leq \frac{c_\ell}{2\epsilon}\text{opt},$$

where we defined $x_\ell = \eta^{L'-\ell(u)}$ and $c_\ell = x_\ell^2\exp(-\frac{cx_\ell}{2})$. The last inequality used Claim 9, which holds since all $u$'s in the summation are at the same level.

Taking the sum over all $\ell$ from 0 to $L'$, we obtain

$$f\sum_{u \in U^*}\Pr[u \in M^1] \leq \frac{\text{opt}}{2\epsilon} \cdot \sum_{\ell=0}^{L'-1}c_\ell = \frac{\text{opt}}{2\epsilon} \cdot \sum_{\ell=0}^{L'-1}x_\ell^2\exp(-\frac{cx_\ell}{2}).$$

Notice that $\{x_\ell : \ell \in [0, L' - 1]\}$ is exactly $\{\eta, \eta^2, \cdots, \eta^{L'}\}$. It is easy to see summation is bounded by a constant for any constant $c$. Thus, the above quantity is at most $O(1/\epsilon)\mathrm{opt}$. Therefore, we proved

$$f \cdot \mathbb{E}\left[|\text{min-set}(M^1 \cap V^\circ)| - |\text{min-set}(M'^0 \cap V^\circ)|\right] \le O(1/\epsilon) \cdot \mathrm{opt}.$$

Notice that Lemma 11 says that $f \cdot |\text{min-set}(M'^0 \cap V^\circ)| \le O(1/\epsilon)\mathrm{opt}$. Thus $f \cdot \mathbb{E}[|\text{min-set}(M^1 \cap V^\circ)|] \le O(1/\epsilon)\mathrm{opt}$.

Then we take vertices outside $V^\circ$ into consideration. Let $U = \text{min-set}(M^1 \cap V^\circ)$. Then $S^1 \subseteq R^1 = \text{min-set}(U \cup (V_T \setminus V^\circ))$. To bound the facility cost of $S^1$, we start with the set $U' = U$ and add vertices in $V_T \setminus V^\circ$ (these are vertices $u$ with $N_u = 0$) to $U'$ one by one and see how this changes $\text{min-set}(U')$. By Claim 16, adding a vertex $N_v = 0$ to $U'$ will either not change $\text{min-set}(U')$, or add $v$ to $\text{min-set}(U')$, or replace an ancestor of $v$ with $v$. In all the cases, the set $\text{min-set}(U') \cap V^\circ$ can only shrink. Thus, we have $R^1 \cap V^\circ \subseteq \text{min-set}(U) = \text{min-set}(M^1 \cap V^\circ)$. We have $E[|R^1 \cap V| \cdot f] \le O(1/\epsilon) \cdot \mathrm{opt}$.

Thus, it suffices to bound the expectation of $|S \setminus V^\circ| \cdot f$. Focus on some $u \in V_T$ with $N_u = 0$. Notice that $u \notin S$ if $\ell(u) \ge L'$. So, we assume $\ell(u) < L'$. In this case there is some leaf $v \in V$ with $N_v > 0$ such that $u$ is the closest point in $R$ to $v$. So $v$ is not a descendant of $u$. Let $u'$ be the ancestor of $v$ that is at the same level at $u$ and define $\pi(u) = u'$. Then $\ell(\pi(u)) = \ell(u)$. Moreover, $u$ is also the closest point in $R$ to $u'$, implying that $\pi$ is an injection. For every $u$, we can bound $f$ as in (3), but with $B_u$ replaced by $B_{\pi(u)}$. Then the above analysis still works since we have $\sum_{u:N_u=0,\ell(u)=\ell} B_{\pi(u)} \le \mathrm{opt}$ for every $\ell \in [0, L' - 1]$ by Claim 9. $\qquad\square$

#### 4.4.2 Increase of connection cost due to the noise

Now we switch gear to consider the change of connection cost due to the noise.

*Proof of Lemma 15.* Focus on a vertex $v$ at level $\ell$ and suppose $v \in S^0$ and some clients are connected to $v$ in the solution produced by Algorithm 2. So, we have $N_v \ge \frac{f}{\lambda^\ell}$. Let the ancestor of $v$ (including $v$ itself) be $v_0 = v, v_1, v_2, \cdots$ from the bottom to the top. Then the probability that $v_0 \notin M^0$ is at most $1/2$ and in that case the connection cost increases by a factor of $\lambda$. The probability that $v_0, v_1 \notin M^0$ is at most $1/4$, and in that case the cost increases by a factor of $\lambda^2$ and so on. As a result, the expected scaling factor for the connection cost due to the noise is at most

$$\sum_{i=0}^{\infty} \frac{1}{2^i} \cdot \lambda^i = \sum_{i=1}^{\infty} \left(\frac{\lambda}{2}\right)^i = O(1).$$

Thus, the connection cost of the solution $S^1$ is at most a constant times that of $S^0$. This is the place where we require $\lambda < 2$. $\qquad\square$

## 5 Lower Bound of UFL for HST Metric

In this section, we prove an $\Omega(1/\sqrt{\epsilon})$ lower bound on the approximation ratio of any algorithm for UFL in the super-set setting under the $\epsilon$-DP model. The metric we are using is the uniform star-metric: the shortest-path metric of a star where all edges have the same length. We call the number of edges in the star its size and the length of these edges its *radius*. By splitting edges, we can easily see that the metric is a $\lambda$-HST metric for a $\lambda > 1$, if the radius is $\frac{\lambda^L}{\lambda-1}$ for some integer $L$.

The main theorem we are going to prove is the following:

**Theorem 17.** There is a constant $c > 0$ such that the following holds. For any small enough $\epsilon < 1, f > 0$ and sufficiently large integer $n$ that depends on $\epsilon$, there exists a set of UFL instances $\{(V, d, f, \vec{N})\}_{\vec{N}}$, where $(V, d)$ is the uniform-star metric of size $n$ and radius $\sqrt{\epsilon}f$, and every instance in the set has $n \le |\vec{N}|_1 \le n/\epsilon$, such that the following holds: no $\epsilon$-DP algorithm under the super-set setting can achieve $c\frac{1}{\sqrt{\epsilon}}$-approximation for all the instances in the set.

*Proof.* Throughout the proof, we let $m = 1/\epsilon$ and we assume $m$ is an integer. We prove Theorem 17 in two steps, first we show the lower bound on an instance with a 2-point metric, but non-uniform facility costs. Then we make the facility costs uniform by combining multiple copies of the 2-point metric into a star metric.

Consider the instance shown in Figure 1a where $V = \{a, b\}$ and $d(a, b) = \sqrt{\epsilon}f$. The facility costs for $a$ and $b$ are respectively $f$ and $0$. Thus, we can assume the facility $b$ is always open. All the clients are at $a$, and the number $N$ of clients is promised to be an integer between $1$ and $m$. We show that for this instance, no $\epsilon$-DP algorithm in the super-set output setting can distinguish between the case where $N = 1$ and that $N = m$ with constant probability; this will establish the $\Omega(\sqrt{m})$ lower bound.

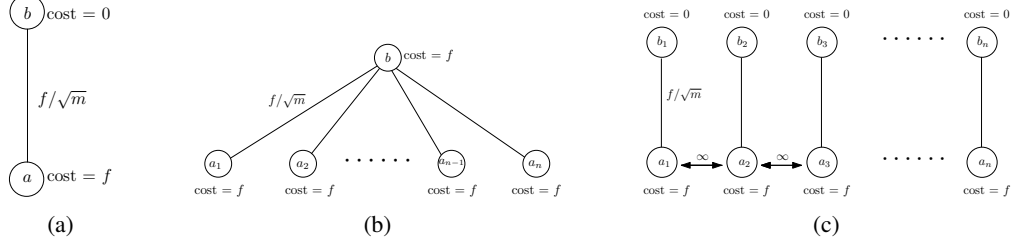

Figure 1: Instance for the lower bound.

Obviously, there are only 2 solutions for any instance in the setting: either we open $a$, or we do not. Since we are promised there is at least 1 client, the central curator has to reveal whether we open $a$ or not, even in the super-set output setting: If we do not open $a$, then we should not include $a$ in the returned set $R$ since otherwise the client will think it is connected to $a$; if we open $a$, then we need to include it in the returned set $R$ since all the clients need to be connected to $a$.

Let $D_i$ be the scenario where we have $N = i$ clients at $a$, where $i \in [m]$. Then the cost of the two solutions for the two scenarios $D_1$ and $D_m$ are listed in the following table:

|       | not open $a$ | open $a$ |
|-------|--------------|----------|
| $D_1$ | $f/\sqrt{m}$ | $f$      |
| $D_m$ | $\sqrt{m}f$  | $f$      |

Thus, if the data set is $D_1$, we should not open $a$; if we opened, we'll lose a factor of $\sqrt{m}$. If the data set is $D_m$, then we should open $a$; if we did not open, then we also lose a factor of $\sqrt{m}$.

Now consider any $\epsilon$-DP algorithm $\mathcal{A}$. Assume towards the contradiction that $\mathcal{A}$ achieves $0.2\sqrt{m}$ approximation ratio. Then, under the data set $D_1$, $\mathcal{A}$ should choose not to open $a$ with probability at least $0.8$. By the $\epsilon$-DP property, under the data set $D_m$, $\mathcal{A}$ shall choose not to open $a$ with probability at least $0.8e^{-(m-1)\epsilon} > 0.8/e \geq 0.2$. Then under the data set $D_m$, the approximation ratio of $\mathcal{A}$ is more than $0.2\sqrt{m}$, leading to a contradiction.

Indeed, later we need an average version of the lower bound as follows:

$$\frac{\sqrt{m}}{\sqrt{m}+1} \, \mathbb{E} \cos t(\mathcal{A}(1); 1) + \frac{1}{\sqrt{m}+1} \, \mathbb{E} \cos t(\mathcal{A}(m); m) \geq cf, \tag{4}$$

where $c$ is an absolute constant, $\mathcal{A}(N)$ is the solution output by the algorithm $\mathcal{A}$ when there are $N$ clients at $a$, and $\cos t(\mathcal{A}(N); N)$ is the cost of the solution under the input $N$. Our argument above showed that either $\mathbb{E} \cos t(\mathcal{A}(1); 1) \geq \Omega(0.2\sqrt{m}) \cdot f/\sqrt{m} = 0.2f$, or $\mathbb{E} \cos t(\mathcal{A}(N); N) \geq 0.2\sqrt{m} \cdot f = 0.2\sqrt{m}f$. In either case, the left side of (4) is at least $\frac{0.2\sqrt{m}f}{\sqrt{m}+1} \geq cf$ if $c$ is small.

The above proof almost proved Theorem 17 except that we need to place a free open facility at location $b$. To make the facility costs uniform, we can make multiple copies of the locations $a$, while only keeping one location $b$; this is exactly a star metric (see Figure 1b). The costs for all facilities are $f$. However, since there are so many copies of $a$, the cost of $f$ for opening a facility at $b$ is so small and thus can be ignored. Then, the instance essentially becomes many separate copies of the 2-point instances we described (see Figure 1c).

However, proving that the "parallel repetition" instance in Figure 1c has the same lower bound as the original two-point instance is not so straightforward. Intuitively, we can imagine that the central curator should treat all copies independently: the input data for one copy should not affect the decisions we make for the other copies. However, it is tricky to prove this. Instead, we prove Theorem 17 directly by defining a distribution over all possible instances and argue that an $\epsilon$-DP algorithm must be bad on average.

Due to the page limit, the detailed analysis is left to the full vesion of the paper.

$\square$

## Acknowledgements

Yunus Esencayi is supported in part by NSF grants CCF-1566356. Part of the work was done when Di Wang and Marco Gaboardi were visiting the Simons Institute of the Theory for Computing. Marco Gaboardi is supported in part by NSF through grant CCF-1718220. Di Wang is supported in part by NSF through grant CCF-1716400. Shi Li is supported in part by NSF grants CCF-1566356, CCF-1717138 and CCF-1844890.

## Footnotes

[2]If this is not the case, we can repeat the following process many steps until the condition holds: create a new root for $T$ and let the old root be its child.

[3]This means for every two distinct vertices $u, v \in V'$, $u$ is not an ancestor of $v$

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
