[Supplementary Material]

# Facility Location Problem in Differential Privacy Model Revisited: Supplementary Material

## A  Connection of our setting to the Joint Differential Privacy model

The *Joint Differential Privacy* model was initially proposed in [Kears-Pai-Roth-Ullman, ITCS 2014]. In the model, every client gets its own output from the central curator and the algorithm is $\epsilon$-joint differentially private (JDP) if for every two datasets $D$, $D'$ with $D' = D \uplus \{j\}$, the joint distribution of the outputs for all clients *except* $j$ under the data $D$ is not much different from that under the dataset $D'$ (using a definition similar to that of the $\epsilon$-Differential Privacy). In other words, $j$'s own output should not be considered when we talk about the privacy for $j$.

The super-set output setting in the $\epsilon$-DP model we considered for UFL is closely related to the JDP-model. In order for the JDP model to be meaningful, one needs to define the problem in such a way that the algorithm needs to give an output for each client, which contains important information for the client. In the UFL problem, this information can be which facility the client should be connected to. If we define UFL in this way, i.e, every client only needs to know its own connecting facility, then our $\epsilon$-DP algorithm in the super-set output setting implies an $\epsilon$-JDP algorithm: Instead of outputting the superset $R$ to everyone, the central curator can simply output to each $j$ the facility that $j$ is connecting to.

## B  Reducing general metrics to HST metrics: Proof of Theorem 1

This section is dedicated to the proof of Theorem 1. The FRT tree decomposition result states that any $n$-point metric can be embedded into a distribution of $O(1)$-HST metrics:

**Theorem I.** There is an efficient randomized algorithm that, given a constant $\lambda > 1$, a metric $(V, d)$ with $n := |V|$, minimum non-zero distance at least 1 and diameter $\Delta := \max_{u,v \in V} d(u, v)$, outputs a $\lambda$-HST $T$ whose leaves are exactly $V$ satisfying the following conditions:

(Ia)  With probability 1, for every $u, v \in V$, we have $d_T(u, v) \geq d(u, v)$.

(Ib)  For every $u, v \in V$, we have

$$\mathbb{E}\, d_T(u, v) \leq O(\log n) \cdot d(u, v),$$

where the expectation is over all randomness of the algorithm.

(Ia) says that the metric $d_T$ restricted to $V$ is always *non-contracting* compared to the metric $d$. On the other hand (Ib) says that $d_T$ "expands" only by a factor of $O(\log n)$ in expectation. Theorem I allows us to reduce many combinatorial problems with min-sum objective on general metrics to those on metrics induced by HSTs. We use the UFL problem as an in this proof.

Let $\lambda > 1$ be a small enough constant as in Theorem 1. Given the UFL instance $(V, d, f, \vec{N})$, our algorithm shall randomly generate a $\lambda$-HST $T$ for the input metric $(V, d)$ using the algorithm in Theorem I. Then we solve the instance $(V, d_T|_V, f, \vec{N})$ using the algorithm $\mathcal{A}$ to obtain a set $S$ of

open facilities.[1] One show that the final approximation ratio we obtain for the original instance $(V, d, f, \vec{N})$ will be $O(\log n)$ times that for the instance $(V, d_T|_V, f, \vec{N})$; moreover the $\epsilon$-DP property carries over in the reduction.

**Claim I.** The expected cost of $S$ to the original instance $(V, d, f, \vec{N})$ is at most $O(\alpha_{\text{tree}} \cdot \log n)$ times that of the optimum solution of $(V, d, f, \vec{N})$.

*Proof.* Let $S^* \subseteq V$ be the optimum solution to the instance $(V, d, f, \vec{N})$. Then, we have

$$\mathop{\mathbb{E}}_{T} \text{cost}_{d_T}(S^*) \leq O(\log n)\text{cost}_d(S^*).$$

Above, $\text{cost}_{d_T}(S^*)$ is the cost of the solution $S^*$ w.r.t instance $(V, d_T|_V, f, \vec{N})$ and $\text{cost}_d(S^*)$ is the cost of the solution $S^*$ w.r.t $(V, d, f, \vec{N})$. The expectation is over the randomness of $T$. The equation simply follows from property (Ib).

If the algorithm for the instance $(V, d_T|_V, f, \vec{N})$ is an $\alpha_{\text{tree}}$-approximation, then we have

$$\mathbb{E}[\text{cost}_{d_T}(S)|T] \leq \alpha_{\text{tree}} \cdot \text{cost}_{d_T}(S^*),$$

where $S$ is the solution output by the algorithm when the chosen tree is $T$. This holds since the optimum cost to the instance $(V, d_T|_V, f, \vec{N})$ is at most $\text{cost}_{d_T}(S^*)$. Then, using property (Ia), i.e, the metrics are non-contracting, we have $\text{cost}_d(S) \leq \text{cost}_{d_T}(S)$. Thus, we have

$$\mathbb{E}[\text{cost}_d(S)|T] \leq \alpha_{\text{tree}} \cdot \text{cost}_{d_T}(S^*).$$

Taking the expectation over the randomness of $T$, we have

$$\mathop{\mathbb{E}}_{T} \text{cost}_d(S) = \mathop{\mathbb{E}}_{T} \big[\,\mathbb{E}[\text{cost}_d(S)|T]\,\big] \leq \alpha_{\text{tree}} \cdot \mathop{\mathbb{E}}_{T} \text{cost}_{d_T}(S^*) \leq O(\alpha_{\text{tree}} \cdot \log n)\text{cost}_d(S^*).$$

$\square$

For the $\epsilon$-DP property of the algorithm, it suffices to observe that the random variable $T$ is independent of $\vec{N}$, and for any fixed $T$, the algorithm $\mathcal{A}$ on $T$ is $\epsilon$-DP. This finishes the proof of Theorem 1.

## C  Missing Proofs from Section 4

*Proof of Claim 1.* Fix the optimum solution for the instance $(V, d_T, \vec{N})$. We define the cost inside a sub-tree $T_v$ to be the total cost of open facilities in $T_v$, and the connection cost of clients in $T_v$ in the optimum solution. Clearly, $\text{opt}$ is at least the sum of costs inside $T_v$ over all $v \in V'$ since the $\{T_v\}_{v \in V'}$ are disjoint. On the other hand, the cost inside $T_v$ is at least $B_v = \min\{f, N_v\lambda^{\ell(v)}\}$: If $T_v$ contains some open facility, then the cost is at least $f$; otherwise, all clients in $T_v$ has to be connected to outside $T_v$, incurring a cost of at least $N_v\lambda^{\ell(v)}$. The claim then follows. $\square$

*Proof of Claim 2.* We prove that any $u \in S$ also has $u \in \text{min-set}(V^\circ \cap M)$. Notice that any $u$ with $\ell(u) > L'$ will not be in $R$ (as it will not be minimal-marked) and thus it will not be in $S$. Any $u$ with $\ell(u) < L'$ and $N_u = 0$ will not be marked and thus will not be in $S$.

Now consider the case $\ell(u) \leq L'$ and $N_u \geq 1$. If $u \in S \subseteq R$ then $u$ is minimal-marked in $M$. It must also be minimal-marked in $M \cap V^\circ$ since all vertices in $V_T \setminus V^\circ$ below level $L'$, in particular, verticies in $V_T \setminus V^\circ$ that are descendants of $u$, are not marked. So $u \in \text{min-set}(M \cap V^\circ)$.

Thus, it remains to focus on a vertex $u$ with $\ell(u) = L'$ and $N_u = 0$. Consider any leaf $v$ with $N_v > 0$ and the level-$L'$ ancestor $u'$ of $v$; so $u' \neq u$ since $N_u = 0$. Then $u' \in M$ and there will be some $u'' \in T_{u'}$ such that $u'' \in R$. We have $d(v, u'') < d(v, u)$ and thus clients in $v$ will not be connected to $u$. This holds for any $v$ with $N_v > 0$. So, $u \notin S$. The finishes the proof of the claim. $\square$

*Proof of Lemma 1.* We break the set of $u$'s in $\text{min-set}(V^\circ \cap M)$ into two cases and bound the total cost for each case separately.

(a) $\ell(u) = L'$. By the definition of $V^\circ$, we have $N_u \geq 1$. Then, $f \leq \frac{1}{\epsilon}\lambda^{L'} \leq \frac{1}{\epsilon}N_u\lambda^{\ell(u)}$ and thus $f \leq \frac{1}{\epsilon}B_u$ by the definitions of $L'$ and $B_u$. So, we have the total facility cost of all $u$'s in this case is at most $\frac{1}{\epsilon}\sum_{u \text{ in the case (a)}} B_u \leq \frac{1}{\epsilon} \cdot \text{opt}$. The inequality is by Claim 1, which holds as all $u$'s in the summation are at the same level.

(b) $\ell(u) < L'$. $u$ must minimal-marked, i.e, $u \in R$. Then we have $f \leq N_u\lambda^{\ell(u)}$ and thus $f \leq B_u$. The total cost in this case is at most $\sum_{u \in R} B_u \leq \text{opt}$ by Claim 1, which holds since $R$ does not contain an ancestor-descendant pair.

Thus, we have proved $F \leq (1 + 1/\epsilon)\text{opt}$. Combining the upper bounds for both $C$ and $F$ gives the lemma. $\qquad\square$

*Proof of Lemma 2.* Notice that reducing $R$ to $S$ will not increase the connection cost of any client since we are using the closest-facility rule. Thus, we can pretend the set of open facilities is $R$ instead of $S$. Let us focus the clients located at any leaf $v \in V$. Let $u$ be the level-$L'$ ancestor of $v$; notice that $u$ is marked. If $v$ is marked, then $v \in R$ and thus the connection cost of any client at $v$ is 0. So we can assume $v$ is unmarked. Then there is exactly one maximal-unmarked vertex $u' \neq u$ in the $u$-$v$ path in $T$. Let $u''$ be the parent of $u'$. Then $u''$ is marked and some vertex in $T_{u''}$ will be in $R$. So the connection cost of a client at $v$ is at most $\frac{2}{\lambda - 1}\lambda^{\ell(u'')} = \frac{2\lambda}{\lambda - 1} \cdot \lambda^{\ell(u')}$. The total connection cost of clients is at most

$$\frac{2\lambda}{\lambda - 1} \sum_{u' \in V_T \text{ is maximal-unmarked}} N_{u'} \cdot \lambda^{\ell(u')} = \frac{2\lambda}{\lambda - 1} \sum_{u' \text{ as before}} B_{u'} \leq \frac{2\lambda}{\lambda - 1} \cdot \text{opt}.$$

For every $u'$ in the summation, we have $N_{u'}\lambda^{\ell(u')} < f$ and thus $B_{u'} = N_{u'}\lambda^{\ell(u')}$. So the equality in the above sequence holds. The inequality follows from Claim 1 , as the set of maximal-unmarked vertices does not contain an ancestor-descendant pair. Thus, we proved the lemma. $\qquad\square$

*Proof of Lemma 3.* We focus on two datasets $\vec{N}, \vec{N'} \in \mathbb{Z}_{\geq 0}^V$ satisfying $|\vec{N'} - \vec{N}|_1 = 1$. Let $v$ be the unique leaf with $|N_v - N_v'| = 1$. We show the $\epsilon$-differential privacy between the two distributions of $M$ generated by Algorithm 2 for the two datasets. Since the returned set $R$ is completely decided by $M$, this is sufficient to establish the $\epsilon$-DP property. For two different vertices $u, u' \in V_T$, the event $u \in M$ is independent of the event $v \in M$, since the former only depends on the noise added to $N_u$ and the later only depends that added to $N_{u'}$. Thus, by the composition of differentially private mechanisms, we can focus on the sub-algorithm that only outputs whether $u \in M$, for each $u \in V_T$.

For simplicity, for every $u \in V_T$, let $a_u$ and $a_u'$ respectively indicate if $u \in M$ under the datasets $\vec{N}$ and $\vec{N'}$. If $\ell(u) \geq L'$ then $a_u = a_u' = 1$ always holds. Also, $a_u$ and $a_u'$ have the same distribution if $u$ is not an ancestor of $v$, since $\vec{N}_u = \vec{N}_u'$. That is the sub-algorithm for $u$ is 0-DP between $\vec{N}$ and $\vec{N'}$.

So we can focus on an ancestor $u$ of $v$ with $\ell(u) < L'$. For this $u$ we have $|N_u - N_u'| = 1$. Due to the property of the Laplacian distribution, the sub-algorithm for this $u$ is $\frac{c\eta^{L'+\ell(v)}}{f}$-differentially private between $\vec{N}$ and $\vec{N'}$. Summing the privacy over all such vertices gives the total privacy as

$$c\sum_{\ell=0}^{L'-1} \frac{\eta^{L'+\ell(v)}}{f} = \frac{c\eta^{L'}}{f} \cdot \frac{\eta^{L'} - 1}{\eta - 1} \leq \frac{c}{f(\eta - 1)} \cdot \lambda^{L'} \leq \frac{c}{f(\eta - 1)} \cdot \lambda\epsilon f = \frac{c\eta\epsilon}{\eta - 1} = \epsilon,$$

by the definition of $c$. $\qquad\square$

*Proof of Claim 3.* Notice that $\text{min-set}(M') = \text{min-set}(\text{min-set}(M) \cup \{v\})$, and $\text{min-set}(M)$ does not contain an ancestor-descendant pair. If $v$ is an ancestor of some vertex in $\text{min-set}(M)$, Case (3a) happens. If $v$ is a descendant of some vertex in $\text{min-set}(M)$, then Case (3c) happens. Otherwise, we have case (3b). $\qquad\square$

# D  Lower Bound on Approximation Ratio for $\epsilon$-DP Algorithm: Making Facility Costs Uniform

In this section, we formally describe the star instance we shall construct for proving Theorem 4 . The star is depicted in Figure 1b : the set of nodes are $V = \{b, a_1, a_2, \cdots, a_n\}$ where $b$ is the center and $a_i$'s are the leaves. The distance between $b$ and $a_i$ for $i \in [n]$ is $f/\sqrt{m}$. The facility cost of each node equals to $f$.

We shall make sure that in every location $a_i$, we have at least 1 client. Then, the optimum solution to the instance has cost at least $n \cdot f/\sqrt{m}$. If our $n$ is at least, say $\sqrt{m} = 1/\sqrt{\epsilon}$, then the cost is at least $f$. So, opening the facility at $b$ will only cost $f$ and thus only lose an additive factor of 1 in the approximation ratio. This does not affect our analysis since we are interested in whether there is a $c\sqrt{m}$-approximation or not. Thus, without loss of generality we can assume the facility at $b$ is open for free. With this assumption, we can see that the instance in Figure 1b is equivalent to instance in Figure 1c : it can be seem as $n$ copies of the instance in Figure 1a , that is for each $i \in [n]$, the facility cost at $b_i$ is 0, while it is $f$ at $a_i$. The distance between $b_i$ and $a_i$ is $f/\sqrt{m}$. Moreover, the distances between the copies are $\infty$. In the following we will focus on instance in Figure 1c .

We can use a vector $\vec{N} \in \mathbb{Z}_{\geq 0}^n$ to denote an input vector, where $N_i$ is the number of clients at location $a_i$. We shall only be interested in the datasets $\vec{N} \in \{1, m\}^n$, i.e, every location has either 1 or $m$ clients. We define a distribution $\mathcal{P}$ over the set of vectors as follows: for every $i \in [n]$, let $N_i = 1$ with probability $\frac{\sqrt{m}}{\sqrt{m}+1}$ and $N_i = m$ with probability $\frac{1}{\sqrt{m}+1}$; the choices for all $i \in [n]$ are made independently.

We consider the term of $\mathbb{E}_{\vec{N} \sim \mathcal{P}}[\text{cost}(S^*(\vec{N}); \vec{N})]$, where $S^*(\vec{N})$ is the optimal solution for the dataset $\vec{N}$. Then, for the optimum solution is easy to define: if $N_i = 1$, we do not open $a_i$ and if $N_i = m$, we open $a_i$. The expected cost of the optimum solution over all vector $\vec{N}$ is

$$\mathbb{E}_{\vec{N} \sim \mathcal{P}}[\text{cost}(S^*(\vec{N}); \vec{N})] = \sum_{\vec{N} \in \{1,m\}^n} P_{\vec{N}'}[\text{cost}(S^*(\vec{N}); \vec{N})] = \sum_{\vec{N} \in \{1,m\}^n} P_{\vec{N}} \sum_{i=1}^{n} \text{opt}(N_i)$$

$$= \sum_{i=1}^{n} \sum_{\vec{N}, \vec{N}' \in \{1,m\}^n, \vec{N}, \vec{N}' \text{ differ at } i, N_i = 1, N_i' = m} [P_{\vec{N}} \cdot \text{opt}(1) + P_{\vec{N}'} \cdot \text{opt}(m)]$$

$$= \sum_{i, \vec{N}, \vec{N}' \text{ as before}} \frac{2 P_{\vec{N}} f}{\sqrt{m}}. \tag{i}$$

Above $\text{opt}(1) = f/\sqrt{m}$ and $\text{opt}(m) = f$ are the optimum cost for the two-point instance when there are 1 and $m$ clients respectively. $P_{\vec{N}}$ is the probability of $\vec{N}$ according to $\mathcal{P}$. The second equality in (i) is due to the structure of instance of Figure 1c and thus $\text{cost}(S^*(\vec{N}); \vec{N}) = \sum_{i=1}^{n} \text{opt}(N_i)$. Notice that we could have a cleaner form for the expected cost; however we use the above form for the purpose of easy comparison.

Now we consider any $\epsilon$-DP algorithm $\mathcal{A}$. Due to the structure of Figure 1c, if we denote the term $\text{cost}_i(\mathcal{A}(N_i); N_i)$ as the cost of $\mathcal{A}(\vec{N})$ given $\vec{N}$ incurred by clients at $a_i$, the expected cost of $\mathcal{A}$ can be written as

$$\mathbb{E}_{\vec{N} \sim \mathcal{P}, \mathcal{A}}[\text{cost}(\mathcal{A}(\vec{N}); \vec{N})]$$

$$= \sum_{i=1}^{n} \sum_{\vec{N}, \vec{N}' \in \{1,m\}^n, \vec{N}, \vec{N}' \text{ differ at } i, N_i = 1, N_i' = m} [P_{\vec{N}} \mathbb{E}\text{cost}_i(\mathcal{A}(\vec{N}); \vec{N}) + P_{\vec{N}'} \mathbb{E}\text{cost}_i(\mathcal{A}(\vec{N}'); \vec{N}')].$$

From the analysis of the two-point metric, we already proved that for any such pair $(\vec{N}, \vec{N}')$ in the summation, we have

$$P_{\vec{N}} \mathbb{E}\text{cost}_i(\vec{N}; \mathcal{A}(\vec{N})) + P_{\vec{N}'} \mathbb{E}\text{cost}_i(\vec{N}'; \mathcal{A}(\vec{N}')) \geq (P_{\vec{N}} + P_{\vec{N}'})cf \geq P_{\vec{N}}cf. \tag{ii}$$

Indeed, if the above property does not hold, we can just use our algorithm $\mathcal{A}$ as a black box to solve the two-point instance: For every $j \neq i$, we just pretend we have $N_j = N_j'$ clients at $a_j$; the number

142 of clients at $a_i$ is exactly the number of clients in $a$ in the two point solution; Run the algorithm $\mathcal{A}$ on
143 the instance. Then the negation of (ii) will imply the negation of (4). Thus (ii) must hold.

144 Thus we have

$$\mathbb{E}_{\vec{N} \sim \mathcal{P}}[\text{cost}(\mathcal{A}(\vec{N}); \vec{N})] \geq \sum_{i, \vec{N}, \vec{N}'} cP_{\vec{N}} f. \tag{iii}$$

145 Comparing the inequality with (i) gives that

$$\mathbb{E}_{\vec{N} \sim \mathcal{P}}[\text{cost}(\mathcal{A}(\vec{N}); \vec{N})] \geq (c\sqrt{m}/2)\mathbb{E}_{\vec{N} \sim \mathcal{P}}[\text{cost}(S^*(\vec{N}); \vec{N})].$$

146 Thus, $\mathbb{E}[\text{cost}(\mathcal{A}(\vec{N}); \vec{N})] \geq (c\sqrt{m}/2)\text{cost}(S^*(\vec{N}); \vec{N})$ holds for at least one vector $\vec{N}$ in the support
147 of $\mathcal{P}$. So the algorithm $\mathcal{A}$ must have an $\Omega(\sqrt{m})$-approximation factor. This finishes the proof of
148 Theorem 4.

## Footnotes

[1] Recall that in the super-set output setting, $S$ is not the returned set, but the set of facilities that are connected by at least 1 client. One small caveat is that for the original instance $(V, d, f, \vec{N})$, given a set $R$ of facilities returned by the algorithm, we should use the tree metric $d_T$ to decide how to connect the clients, instead of the original metric $d$. Thus, along with the set $R$, the algorithm should also return the HST $T$.