[Reviews · NeurIPS 2019]

Reviewer 1



This paper studies the facility location problem subject to a privacy constraint that is a relaxation of differential privacy. Although the paper calls it “super-set output setting”, the privacy model is the same as joint differential privacy (JDP), which is studied in other works usually related to algorithmic game theory. Essentially, the super-set output setting is the “billboard” used to show all the users and then the users can use their own private information to determine their action based on the billboard. Although the main paper does not make this connection explicit, it is covered in the supplementary file. That being said, I think the paper is more interesting when presenting the DP case as too stringent and thus seeing what can be achieved with the known relaxation of JDP. In the introduction, the paper states a nice result that shows more can be achieved with this relaxed privacy model, thus showing a gap between the privacy models. The paper would be significantly stronger if the lower bound 1/\sqrt{\eps} matched the upper bound 1/eps. However, the paper still shows that we can still obtain decent utility using relaxations of differential privacy when DP is too stringent of a condition. However, I am unsure about the accuracy of the upper bound result. In particular, Algorithm 1 and Algorithm 2 do not have epsilon in the pseudocode (just that epsilon is an input). I am unsure how Algorithm 1 would run, as I do not know what L and L’ are (are they fixed levels in the tree?). Is there some condition on L’ or f that ensures Lemma 1? It seems like critical information is not in the paper. I am unable to see how the privacy parameter affects the algorithms, despite it being an input parameter. I did not verify the lower bound analysis. Comments: - Explain in the related work section why previous methods cannot be used. - In the Preliminaries “Also denote … denote” - What are “LDP algorithms" (see beginning of Section 3) - Page 4: “verticies” #### Because of the author feedback I have increased my score from 4 to 5 since the feedback pointed out the dependence of the privacy parameter in the algorithm. Thank you for that! The paper would significantly benefit from a thorough revision of typos and clearly stating the dependencies of various parameters. I also think the work would be benefit from framing the privacy analysis around Joint Differential Privacy, since it could then be placed in that line of work. At the very least the similarity in their setting to JDP should not be pushed to the supplementary material.

Reviewer 2



originality: novel use of HST metric embedding and algorithm specialized to HST metrics. clarity: The overall thrust of the paper is clear. however, the grammar and writing is consistently slightly wrong throughout. significance: these results significantly improve SOTA for the facility location problem under the constraint of differential privacy. comment: * why frame this problem as the super output setting instead of framing it as an allocation problem subject to Joint Differential Privacy (JDP), since that is how the solution is framed.

Reviewer 3



The paper improves the approximation ratio of an important problem. The algorithm, however, is largely identical to the previous one, except that it keeps only a frontier of marked vertices as facilities rather than all of them. The change does make the analysis more challenging but not overly so. In any case, given that facility location is such a fundamental problem in computer science and it improves to ratio to almost tight (to this end, I'd appreciate the paper much more if the lower bound is proved to be 1/eps), it is a viable acceptance to NeurIPS.

[Author Response · NeurIPS 2019]

We would like to thank all reviewers for their thoughtful comments. Below we address their concerns.

**For Reviewer 2:**    First, we address the major comment of Reviewer 2 regarding where $\epsilon$ is used in the two algorithms. **Indeed, $L'$, which was defined in Line 149 of page 4, depends on $\epsilon$.** So both algorithms depend on $\epsilon$. We are sorry for the confusion. In the final version, we shall define $L'$ in the pseudo-codes to emphasize the dependence on $\epsilon$.

To give more details, $L'$ is the lowest level $\ell$ in $T$ satisfying $\lambda^\ell \geq \epsilon f$. Note that in both algorithms, we are marking all vertices on level $L'$ and this means there cannot be any minimal-marked vertices above level $L'$. Thus, the open facilities can only be at level $L'$ or at lower levels. In the analysis of the total facility cost obtained in Algorithm 1, which is the proof of lemma 1, we obtain an upper bound $\frac{1}{\epsilon} \cdot$ opt for the cost of open facilities which are at level $L'$, using $f \leq \frac{1}{\epsilon}\lambda^{L'}$. Combining with the facilities at lower levels, we upper bound the total facility cost with $(1 + \frac{1}{\epsilon})$opt. Similarly, in the proof of lemma 4, in finding an upper bound for the total expected facility cost of Algorithm 2, we are using $\lambda^{L'} \geq \epsilon f$ property of $L'$. This is how the privacy parameter $\epsilon$ comes out in the analysis of our algorithms.

Explain in the related work section why previous methods cannot be used.

As discussed in our paper, the techniques of Gupta et al. [8] can give an $\epsilon$-DP algorithm for UFL under our model, but with much worse approximation ratio. We have many new ingredients in our paper that led to the improved approximation ratio.

What are "LDP algorithms" (see beginning of Section 3)

This is a typo. LDP means "locally differentially private". We derived an LDP algorithm for the problem, but did not include it in the paper.

We will fix typos and add additional explanations in the final version. Thanks for pointing them out.

**For Reviewer 3:**    Why frame this problem as the super output setting instead of framing it as an allocation problem subject to Joint Differential Privacy (JDP), since that is how the solution is framed.

We used the super-set output setting because this describe more precisely how the solution is presented. In our algorithm, we are not revealing each user the location to which they connect, instead we are revealing a super-set of the open facilities and each client connects to the closest facility in the set. Of course this is very related to JDP as discussed in Section A of the supplementary material. But using the super-set is just one way to guarantee the JDP property. We will discuss our terminology more thoroughly in the final version.

[Meta-Review · NeurIPS 2019]

This paper deals with the differentially private facility location problem. The facility location problem is a Lagrangian formulation of the metric clustering problem. Given a set of points in a metric space (for example Euclidean space), the goal is to find a set of cluster centers that minimizes the sum of distances from the points to their closest cluster center, plus a regularization parameter times the number of centers. Traditionally, this problem has attracted a lot of attention in the algorithms community, given that it captures well the clustering problem, and tools developed for this problem have transferred well to several other formulations of clustering. This work studies the differentially private version of this clustering problem. This problem is studied in a version of privacy that has been studied for this problem, known as Joint Differential Privacy. The previous work for this problem derived an polylog approximation to this problem, by reducing the problem to a class of metrics known as HSTs, and deriving a polylog approximation ratio for that problem. The current work improves the approximation factor to O(1) for HSTs and O(log n) for arbitrary metric spaces. Further the author give the first lower bounds for this problem. The algorithm builds on the previous work of Gupta et al., and shows that few small changes to the algorithm, and more sophisticated analysis, one can get an O(1) approximation (for constant privacy parameter) on HSTs. The reviewers agreed that this is a fundamental combinatorial optimization problem, variants of which arrive in clustering problems in ML. Given the importance of private Machine Learning algorithms, new tools and analyses are of interest. This paper makes significant progress on this important question. The reviewers would be happier with the lower bound matching in dependence on epsilon. They also hope that the writing clarity will be improved based on the feedback. Overall, the technical novelty and the importance of the question convinced the reviewers and the meta-reviewer to support acceptance for this work.